# Evaluating the prevalence and risk factors for macrolide resistance in *Mycoplasma genitalium* using a newly developed qPCR assay

Joyce F. Braam[1]*, David J. Hetem[2], Clarissa E. Vergunst[1,3], Sophie Kuizenga Wessel[4], Martijn S. van Rooijen[1], Roel H. T. Nijhuis[2,5], Maarten F. Schim van der Loeff[1,6], Alje P. van Dam[1,7], Sylvia M. Bruisten[1,7]

1 Department of Infectious Diseases, Public Health Service (GGD) of Amsterdam, Amsterdam, The Netherlands, 2 Department of Medical Microbiology, Haaglanden Medical Center, The Hague, The Netherlands, 3 Department of Dermatology, NWZ Den Helder, Den Helder, The Netherlands, 4 Department of Infectious diseases, Public Health Service, The Hague, The Netherlands, 5 Laboratory for Medical microbiology and immunology, Meander Medical Center, Amersfoort, The Netherlands, 6 Department of Internal Medicine, Amsterdam Infection & Immunity Institute (AI&II), Amsterdam University Medical Center (UMC), Amsterdam, The Netherlands, 7 Department of Medical Microbiology, Amsterdam Infection & Immunity Institute (AI&II), Amsterdam University Medical Center (UMC), Amsterdam, The Netherlands

* jbraam@ggd.amsterdam.nl

**Data Availability Statement:** All relevant data are within the manuscript and its Supporting Information files.

## Abstract

*Mycoplasma genitalium* (MG) is a sexually transmitted bacterium in which macrolide resistance is rapidly increasing, limiting treatment options. We validated a new assay to detect the presence of macrolide resistance-associated mutations in MG (MG-MRAM). In 2018, symptomatic and asymptomatic clients visiting sexually transmitted infections (STI) clinics in Amsterdam or The Hague were tested for MG using transcription mediated amplification (TMA) assays. The sensitivity to detect MG of the newly developed MG-MRAM qPCR was compared to the MgPa qPCR, both in relation to the TMA assay. For the sensitivity and specificity to detect relevant mutations the MG-MRAM qPCR was compared to 23SrRNA sequencing analysis. The qPCR was subsequently used to determine the presence of MG-MRAM at different anatomical locations and to identify risk factors for MG-MRAM. MG-positive clients (402) providing 493 MG-positive samples were included. In total 309/493 (62.7%) samples from 291 (72.4%) clients were successfully typed with the MG-MRAM qPCR. The MG-MRAM qPCR had a sensitivity of 98.6% (95%CI 91.1%-99.9%) and specificity of 94.1% (95%CI 78.9%-99.0%) to detect MG-MRAM compared to sequencing analysis. Infection with MG-MRAM was detected in 193/291 (66.3%) clients: in 129/178 (72.5%) men and 64/113 (56.6%) women (p = 0.005). Prevalence of MG-MRAM was significantly higher in men, clients with a higher education, HIV-positive clients and clients with >10 sexual partners in the previous six months, but in multivariable analysis no factor was significantly associated with MG-MRAM presence. Since MG-MRAM prevalence was very high, testing for MG-MRAM is essential if treatment for MG is considered, and can be performed with this sensitive and specific qPCR test in routine diagnostics.

**Funding:** For this research, the authors received part of the diagnostic tests to detect MG from Hologic Inc, San Diego, USA. The funder had no role in study design, data collection and analysis, decision to publish, or preparation of the manuscript.

**Competing interests:** The authors have read the journal's policy and have the following potential competing interests: Hologic Inc, San Diego, USA provided part of the diagnostic tests to detect MG. This does not alter our adherence to PLOS ONE policies on sharing data and materials. There are no patents, products in development or marketed products associated with this research to declare.

## Introduction

*Mycoplasma genitalium* (MG) is a sexually transmitted organism which infects 3.1–4.5% of Dutch clients undergoing screening for sexually transmitted infections (STI) [1, 2]. Among men, the most common clinical manifestation of MG infection is non-gonococcal urethritis (NGU). A meta-analysis from 2011 showed that MG was strongly associated with NGU (pooled OR 5.5 [95% CI: 4.4–7.0]) [3]. In women MG has been associated with an increased risk of cervicitis, pelvic inflammatory disease, preterm birth, and spontaneous abortion [4].

According to European guidelines, uncomplicated MG infections should be treated with azithromycin 500 mg PO on day one followed by 250 mg on days 2–5 [5]. However, single dose treatment (1000 mg) with azithromycin is often the preferred treatment of NGU in many countries, including the Netherlands [6]. *Chlamydia trachomatis* (CT) infections are also treated with 1000 mg azithromycin, but often without excluding co-infection with MG [6].

A meta-analysis indicated that a single dose of azithromycin facilitates macrolide resistance in MG [7], but this was not confirmed in a more recent extensive retrospective study in Australia [8]. Macrolide resistance is rapidly increasing worldwide and may be found in up to 89% of clients with urethritis [9]. In the Netherlands, macrolide resistance ranges between 20.9–44.4% [10–12]. Macrolide resistant MG can be treated with moxifloxacin 400 mg PO for 7–10 days [5].

Dutch STI guidelines do not recommend routine testing for MG [6, 13]. In the recent revision, screening for MG in men with NGU is mentioned, but not explicitly advised. Previously several qPCRs have been described with high sensitivity and specificity to detect MRAM [14–18]. Also commercial (CE-IVD cleared) diagnostic tests for macrolide resistance-associated mutations (MRAM) have become available [17]. These tests detect several mutations that are associated with macrolide resistance in the V-region of the 23S rRNA gene: A2058G, A2058T, A2058C, A2059G, and A2059C (*Escherichia coli* numbering) [10, 19]. However, the RealAccurate TVMGres (Pathofinder) assay does not detect the rare A2059C mutation [17]. Since resistance is rapidly increasing, identifying clients with MG-MRAM using routine diagnostic testing might help targeted treatment. As CE-IVD cleared assays mostly have limited systems and specimen types included in their CE-IVD accreditation, these assays cannot be used in all routine diagnostic settings, without extra investments. Therefore we set up and validated a new assay, which uses locked nucleic acids (LNA) in the probes for the specific detection of mutations in the 23SrRNA gene to detect MRAM in MG (MG-MRAM). Using this test, we determined the prevalence of MG-MRAM at different anatomical locations of persons who visited two Dutch sexually transmitted infections (STI) clinics, and we analyzed risk factors for MG-MRAM.

## Methods

### Sample selection

In this study we used a subset of a large cross-sectional MG prevalence study of two STI outpatient clinics in The Netherlands. In the large cross-sectional study all clients–symptomatic and asymptomatic–at the STI outpatient clinic in Amsterdam in February and March 2018 and in May and June 2018 in The Hague were eligible to be included. A full description of the study is provided by Hetem *et al.*, 2020 (in preparation). Briefly, first-void urine was collected from all males and anal swabs were collected from men who have sex with men (MSM). Cervico-vaginal swabs were taken from all female clients. Anal samples were taken from all females attending the STI clinic in The Hague, whereas in Amsterdam anal samples were taken only from females if they reported anal sex or anal symptoms, were notified for an STI, or reported to

perform sex work–both according to the local STI clinic policy. All samples were collected in Aptima sample collection medium and routinely tested for *Neisseria gonorrhoeae* (NG) and CT and additionally for MG for the present study. Only clients of whom at least one sample tested positive for MG were included in the analyses presented here. Socio-demographic data, sexual behavior and the presence of clinical symptoms–including urogenital discharge, dysuria, ulcers, blood loss and pain–were extracted from electronic patient files at the STI clinics. Only results from the initial STI screening were included.

## Detection of MG, CT and NG

The MG-transcription mediated amplification (TMA) assay (Hologic Inc, San Diego, USA) was used for the detection of MG and the Aptima Combo 2 (AC2) TMA was used for the detection of CT and NG on the Panther system according to the manufacturer's instructions. Samples with equivocal results were retested by using the Aptima CT single assay (Hologic) for CT, and an NG qPCR targeting the *opa* genes for NG [20].

## Deoxyribonucleic acid extraction

Next, deoxyribonucleic acid (DNA) was extracted from all samples that tested positive for MG in the MG-TMA assay. From the original Aptima tubes 200 μL sample was used for DNA isolation by isopropanol precipitation [21]. The DNA pellet was dissolved in 50 μL of 10 mM Tris/HCl pH 7 and stored at -20˚C until used for DNA amplification.

## MG-MRAM testing

DNA isolates were tested for MRAM using a newly designed multiplex qPCR to detect the mutations in the 23S rRNA gene at nucleotide positions 2058 and 2059 (*E. coli* numbering) associated with macrolide resistance. LNA probes have previously been designed to be able to detect point mutations [22]. Therefore, we used LNA probes to obtain a high specificity to detect the most prevalent mutations in the 23S rRNA gene. Primer and probe sequences are shown in Table 1. The assay consists of two multiplex qPCRs which use the same forward and reverse primers, 250 nM of each primer and three probes of 125nM each. The wild type (WT) probe contains a FAM label and all mutation probes contain a HEX label. Multiplex mix 1 contained the probes to detect WT, and mutations A2058G and A2059G. Mix 2 contained probes to detect mutations A2058C, A2058T and A2059C. Each multiplex qPCR was performed using 2 μL input of the DNA solution, 18 μL mix solution containing 10 μL Platinum Q Supermix (Invitrogen, Nieuwerkerk a/d IJssel, Netherlands) and primers and probes as indicated. Amplification was performed on a RotorGene (Qiagen, Hilden, Germany) with the following cycling profile: 2 min 50˚C, denaturation at 95˚C for 2 min, followed by 45 amplification cycles consisting of 95˚C for 15 sec, and annealing and extension at 60˚C for 45 sec. Samples with a cycle value of Ct<37 were considered positive. If the Ct value was between 37 and 45 the test was repeated and deemed positive if Ct<45 with a good S-curve. The performance of the MG-MRAM qPCR was determined by comparing to the TMA assay test results.

## MgPa qPCR testing

The MG-MRAM qPCR was compared for sensitivity with the MgPa qPCR [23]–both in relation to the TMA assay. The MgPa qPCR was performed using 2 μl input of the DNA isolate, in a total volume of 20 μl mix solution containing 10 μl Platinum Q Supermix (Invitrogen, Nieuwerkerk a/d IJssel, Netherlands) and 625 nM of each primer and 75 nM of the FAM-labeled probe (based on Jensen *et al.*, 2004 [23]). Amplification was performed on a Rotorgene with

**Table 1. Primers and probes for 23S rRNA sequencing, and MG-MRAM and MgPa qPCR.**

| | Name | Target/ Mutation | Fluorophore | 5'→3' sequence | Quencher | Mix |
|---|---|---|---|---|---|---|
| **Primers** | MG 23S For | 23S rRNA | | GGT GAA GAC ACC CGT TAG G | | 1,2 |
| | MG 23S Rev | 23S rRNA | | CCT ATT CTC TAC ATG GTG GTG TT | | 1,2 |
| | MG 23S Seq For | 23S rRNA | | ***TGTAAAACGACGGCCAGT*** GAAGGTTAAAGAAGGAGGTTAGCAAT | | Sequencing |
| | MG 23S Seq Rev | 23S rRNA | | ***CAGGAAACAGCTATGACC*** CTACCTATTCTCTACATGGTGGTGTTT | | Sequencing |
| | MgPa For | MgPa | | GAG AAA TAC CTT GAT GGT CAG CAA | | MgPa |
| | MgPa Rev | MgPa | | GTT AAT ATC ATA TAA AGC TCT ACC GTT GTT ATC | | MgPa |
| **Probes** | 23S WT LNA | Wild type | FAM | ACG **GAA** **A**GA C**C**C | IABkFQ | 1 |
| | 23S GA LNA | A2058G | HEX | ACG **GGA** A**G**A CC | IABkFQ | 1 |
| | 23S AG LNA | A2059G | HEX | CG G**AG** **A**G**A** **C**C | IABkFQ | 1 |
| | 23S CA LNA | A2058C | HEX | ACG **GCA** A**G**A CC | IABkFQ | 2 |
| | 23S TA LNA | A2058T | HEX | ACG **GTA** **A**GA C**C**C | IABkFQ | 2 |
| | 23S AC LNA | A2059C | HEX | AC**G** G**AC** **A**GA CC | IABkFQ | 2 |
| | MgPa | | FAM | ACT TTG CAA TCA GAA GGT | MGBNFQ | MgPa |

LNA (locked nucleic acid) nucleotides are shown in bold and underlined. M13 tag in sequencing primers are shown in bold and italic. MG-MRAM: *Mycoplasma genitalium* with macrolide resistance-associated mutations; FAM: green fluorescent label, HEX: yellow fluorescent label; MGBNFQ: Taqman minor groove binder non-fluorescent quencher; IABkFQ: Iowa Black® quencher. Multiplex mix 1 contained the probes to detect WT, and mutations A2058G and A2059G. Mix 2 contained probes to detect mutations A2058C, A2058T and A2059C.

the MG-MRAM qPCR cycling program and the same criteria were used to consider a sample as MG positive. Positive results in the MG-MRAM qPCR in samples that were negative in the MgPa qPCR, as well as positive MgPa qPCR results in samples not reactive in the MG-MRAM qPCR were considered as valid since all samples had tested positive in the MG-TMA assay.

## Sequencing analysis of 23SrRNA gene

To confirm the mutation qPCR test results, a subset of 126 of the 309 samples (40.8%) that were typable with the MG-MRAM qPCR were used to perform sequencing analysis (Fig 1). In addition, also a subset of 33 MG-positive samples out of the total of 184 samples (17.9%) in which no MG was detected with the MG-MRAM qPCR was used for sequencing analysis (Fig 1), totaling 159 samples for sequencing analysis. For both subsets we selected samples that tested positive with the MgPa qPCR and had a Ct value of <36. The 23SrRNA forward and reverse primers were tagged with an M13 sequence for the sequencing qPCR reaction (Table 1). Sequencing reactions were performed using 3nM of each primer. Sequences were analyzed using software packages MEGA (version M6.0.6) and Bionumerics (version 7.6.3). Sensitivity and specificity to detect MG-MRAM was determined by comparing the MG-MRAM qPCR to sequencing analysis.

## Statistical analysis

Data were analyzed using SPSS v 20.0 (IBM Corp, Armonk, NY, USA) and significance was assessed two-sided for all variables, with p<0.05. The variable 'Education' was categorized into low (no education, primary school, lower secondary vocational education and intermediate secondary general education), mid (higher secondary general education, senior secondary vocational education and pre-university secondary education) and high (higher professional or university education). A client was considered to be infected by macrolide susceptible MG when the MG-MRAM qPCR only detected WT variants in all the available samples of that

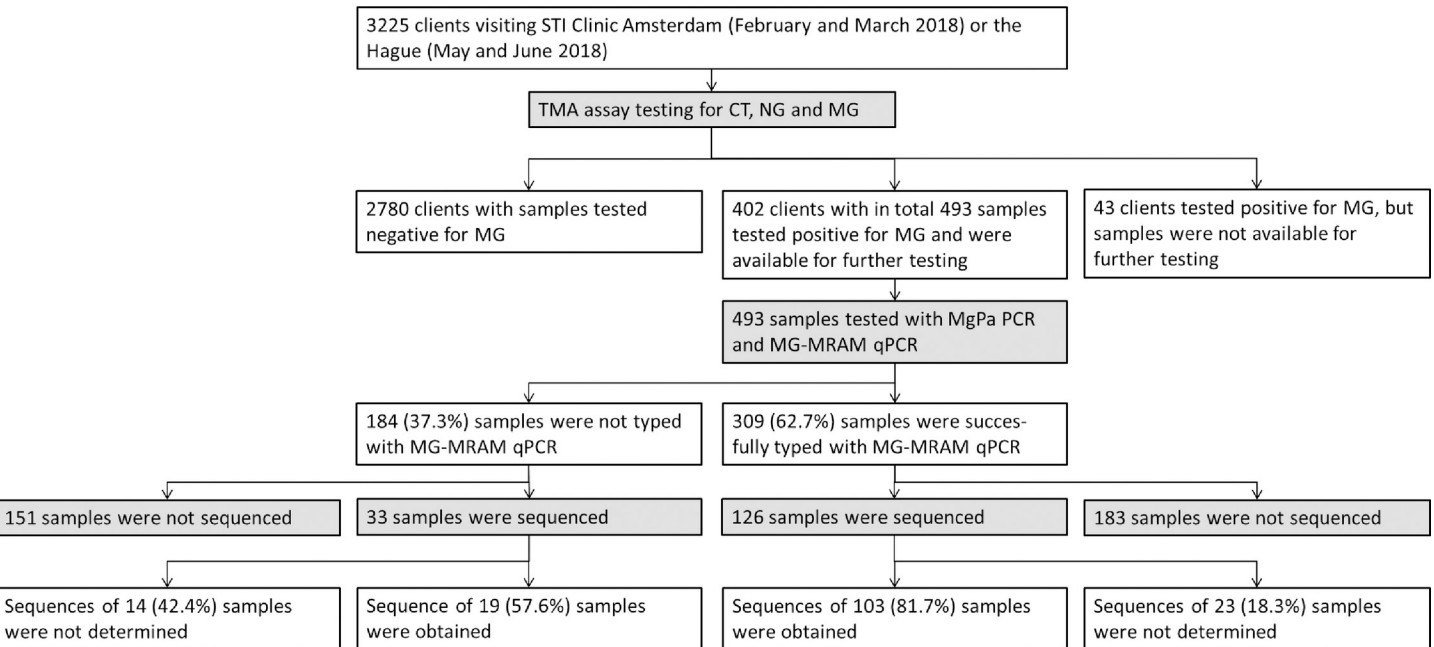

STI: sexually transmitted infections; CT: *Chlamydia trachomatis*, NG: *Neisseria gonorrhoeae*, MG: *Mycoplasma genitalium*, MRAM: macrolide resistance-associated mutations.

**Fig 1. Flow diagram summarizing the number of clients and samples included in this study and which tests were performed on which samples.**

client. If in one of the available samples MG-MRAM was found, the client was considered to be infected by a resistant strain. Univariable analysis was performed using the Chi-square test or Fischer exact test to compare the proportion positive MG-MRAM between sexual risk groups for different anatomical locations. MG-MRAM prevalence was calculated as percentage of typeable samples. Cohen's Kappa was calculated to assess agreement between the MgPa qPCR and MG-MRAM qPCR. Generalized Estimating Equations logistic regression was used to compare the MG-positivity between anatomical locations. Factors associated with MG-MRAM were examined using univariable and multivariable logistic regression analysis on client level.

## Ethics statement

Clients of the STI outpatient clinics were informed of the "opt-out" system regarding research on remnants of client material. Material from clients was only included in this study if clients did not opt-out. All data were fully anonymized before assessment. Results of MG testing were not disclosed to healthcare professionals or clients. The study protocol was evaluated by the Institutional review board which is the Medical Ethics Committee of the Academic Medical Center in Amsterdam (letter reference no. W18.013#18.024) and deemed not to require a full review of the board and informed consent was not deemed to be required.

## Results

### Detection of MG by the MgPa qPCR

During the study period 445/3225 (13.8%) tested clients were positive for MG with the MG-TMA assay, of whom 1031 were MSM, 927 heterosexual men, 1249 women, 17 transgender people and the sexual orientation of 2 patients was unknown. Information on all clients

enrolled during the study period is described by Hetem *et al*. 2020, (manuscript in preparation). From 402/445 clients positive for MG–of whom 182 were MSM, 66 heterosexual men and 154 women–samples were available for MgPa and MG-MRAM testing, including 136 vaginal, 120 urine and 237 anal samples, totaling 493 samples (Fig 1). MG was detected in 99/136 (72.8%), 72/120 (60.0%), and 134/237 (56.5%) of the vaginal, urine and anal samples respectively with the MgPa assay (S1 Table). The MgPa assay detected significantly less often MG in anal samples compared to vaginal samples (p = 0.002) and had a sensitivity of 61.9% (305/493) compared to the TMA assay.

## Detection of MG by the MG-MRAM qPCR

Presence of MG-MRAM or MG-WT could be assessed in 91/136 (66.9%), 72/120 (60.0%) and 146/237 (61.6%) of the vaginal, urine and anal samples respectively by the MG-MRAM qPCR. The MG-MRAM qPCR was negative in 184/493 (37.3%) of the samples that tested positive for MG in the TMA assay (S1 Table). There was no significant difference in detection of MG in vaginal and anal samples (p = 0.307) or vaginal and urine samples (p = 0.253). The MG-MRAM qPCR detected MG in 62.7% (95%CI 58.2%-67.0%) and the MgPa qPCR in 61.9% (95%CI 57.4%-66.1%) of all samples that were positive according to the TMA assay. The overall Cohen's kappa for agreement of the MgPa qPCR and MG-MRAM qPCR for detecting MG was 0.646; stratified per anatomical location kappa was 0.617 for vaginal samples, 0.715 for urine samples and 0.615 for anal samples.

## Sequencing analysis

Sequences were successfully obtained for 103/126 (81.7%) samples that were typed with the MG-MRAM qPCR. One of the 126 typed samples was a mixture of WT and MG-MRAM according to the MG-MRAM qPCR, and thus classified as MG-MRAM. With sequencing analysis that sample was identified as WT. Furthermore, one typed sample was identified as MG-MRAM with MG-MRAM qPCR, but with sequencing as WT; and one sample as WT with MG-MRAM qPCR, but with sequencing analysis as MG-MRAM. The sensitivity of the MG-MRAM qPCR to detect MG-MRAM was 98.6% (95%CI 91.1%-99.9%) and the specificity was 94.1% (95%CI 78.9%-99.0%) (S2 Table). From the 167 MG-positive samples that were not successfully typed with the MG-MRAM qPCR, a subset of 33 were used for sequencing analysis and of those sequences were obtained from 19/33 (57.6%) samples (Fig 1 and Table 2). In these sequenced samples a relative high proportion of A2058T (47.4%) and A2059C (15.8%) was found compared to samples that were sequenced and typed (1.9% and 0%, respectively), but absolute numbers were low for these mutations. Taken together, sequences of 122 MG samples were available in this study with a total of 30.3% wild type and thus 69.7% MG-MRAM types (Table 2). According to sequence analysis the most prevalent mutation was A2059G (32.0%), followed by A2058G (26.2%), A2058T (9.0%) and A2059 C (2.5%) (Table 2).

## Prevalence of MG-MRAM in urogenital and anal samples

MG-MRAM was detected in 65.3% (213/326) of the samples and MG-WT in 34.7% (113/326) of the samples using MG-MRAM qPCR or sequencing analysis (Table 3). Two of 326 typed samples contained both MG-WT and MG-MRAM and were classified as MG-MRAM. The prevalence of MG-MRAM in anal samples was significantly higher in MSM than in women (75.5% vs. 59.2%, p = 0.04) (Table 3). No difference between prevalence of MG-MRAM in MSM (67.6%) and heterosexual males (68.2%) in urine samples was observed (Table 3). From 91 clients samples from two anatomical sites were available of whom 34 (39.5%) MG-MRAM or MG-WT could be successfully typed in both samples. In 31/34 (91.1%) clients the same MG

**Table 2. Type of 23SrRNA mutations in samples from different anatomical locations that were MG positive in the MG-TMA.**

| | Successfully typed and sequenced per anatomical location | | | | | | Sequenced samples that were not typed with | Total successfully sequenced samples[1] |
| --- | --- | --- | --- | --- | --- | --- | --- | --- |
| Sequence | Vagina | Urine hetero-sexual male | Urine MSM | Anus female | Anus MSM | Total | MG-MRAM qPCR | |
| Wild type | 12 (11.7%) | 8 (7.8%) | 3 (2.9%) | 2 (1.9%) | 9 (8.7%) | 34 (33.0%) | 3 (15.8%) | 37 (30.3%) |
| A2059G | 9 (8.7%) | 3 (2.9%) | 6 (5.8%) | 3 (2.9%) | 17 (16.5%) | 38 (36.9%) | 1 (5.3%) | 39 (32.0%) |
| A2058G | 9 (8.7%) | 9 (8.7%) | 4 (3.9%) | 3 (2.9%) | 4 (3.9%) | 29 (28.2%) | 3 (15.8%) | 32 (26.2%) |
| A2058T | 1 (1.0%) | | | | 1 (1.0%) | 2 (1.9%) | 9 (47.4%) | 11 (9.0%) |
| A2059C | | | | | | | 3 (15.8%) | 3 (2.5%) |
| Total | 31 (30.1%) | 20 (19.4%) | 13 (12.6%) | 8 (7.8%) | 31 (30.1%) | 103 (100%) | 19 (100%) | 122 (100%) |

[1] In total 159/493 (32.2%) of the samples were subjected to sequence analysis. Of these 159, 122 (103 +19) yielded useful sequences.

In the left part of the table samples are shown that were typed with the MG-MRAM qPCR and successfully sequenced. In the middle column results are shown of 19/33 successfully sequenced samples that were not typed with the MG-MRAM qPCR. Right column is showing the total number of successfully sequenced samples in this study.

resistance type was found, but in 3/34 (8.8%) clients MG-WT was found at one location and MG-MRAM at the other location. Specifically, one of these clients had anal MG-WT and vaginal MG-MRAM; another had anal MG-MRAM and vaginal MG-WT; and the last one had anal MG-MRAM and urine MG-WT.

## Risk factors for MG-MRAM infection

The study included 402 clients who provided 493 samples. For 291/402 clients (72.4%) one or more of the samples were typeable using the MG-MRAM qPCR or sequencing analysis. If in one of the available samples MG-MRAM was found, the client was considered to be infected by a resistant strain. MG-MRAM was detected in at least one of the samples from 193/291 (66.3%) clients (Table 4). MG-MRAM was more common in men (72.5%, p = 0.005), in clients with higher education (72.6%, p = 0.029), in clients with >10 sexual partners in the preceding six months (77.5%, p = 0.016) and in HIV-positive clients (82.8%, p = 0.047). All HIV-positive

**Table 3. Prevalence of MG with Macrolide Resistance-Associated Mutations (MG-MRAM) in samples derived from different anatomical locations typed by MG-MRAM qPCR and/or by sequencing analysis.**

| Location | | Total (MG-TMA positive) | Typeable[2] (% of total) | MG-MRAM (% of typeable) | P- value |
| --- | --- | --- | --- | --- | --- |
| Vagina | Women | 136 | 97 (71.3%) | 54 (55.7%) | |
| Urine | Total | 120 | 78 (65.0%) | 53 (67.9%) | |
| | MSM | 54 | 34 (63.0%) | 23 (67.6%) | 0.960 |
| | Heterosexual male | 66 | 44 (66.7%) | 30 (68.2%) | |
| Anus | Total | 237 | 151 (63.7%) | 106 (70.2%) | |
| | MSM | 147 | 102 (69.4%) | 77 (75.5%) | 0.040 |
| | Women | 90 | 49 (54.4%) | 29 (59.2%) | |
| Total[1] | | 493 | 326 (66.1%) | 213 (65.3%) | |

MSM: men who have sex with men.

[1] From 91 clients samples from two different anatomical locations were available.

[2] Typeable included data from the MG-MRAM qPCR combined with sequencing data; 17/326 (5.2%) samples could only be typed using sequencing analysis, but not with the MG-MRAM qPCR.

Prevalence of MG-MRAM was defined as proportion of typeable samples.

**Table 4. Prevalence of MG-MRAM among clients of the STI outpatient clinic in Amsterdam and The Hague according to different client characteristics, and results of logistic regression analysis for association of characteristics with presence of MG-MRAM.**

| | | Total clients | Typeable (% of total) | MG-MRAM (% of typeable) | P-value[2] | OR | 95%CI | AOR[3] | 95%CI |
|---|---|---|---|---|---|---|---|---|---|
| Overall[1] | | 402 | 291 (72.4%) | 193 (66.3%) | | | | | |
| Sex | Women | 154 | 113 (73.4%) | 64 (56.6%) | 0.005 | 1 | | | |
| | Men | 248 | 178 (71.8%) | 129 (72.5%) | | 2.02 | 1.23–3.31 | | |
| Sexual risk group | Women | 154 | 113 (73.4%) | 64 (56.6%) | 0.016 | 1 | | 1 | |
| | Heterosexual male | 66 | 44 (66.7%) | 30 (68.2%) | | 1.64 | 0.79–3.42 | 1.38 | 0.73–2.60 |
| | MSM | 182 | 134 (73.6%) | 99 (73.9%) | | 2.17 | 1.27–3.70 | 1.66 | 0.77–3.61 |
| Age in years | <25 | 160 | 115 (71.8%) | 69 (60.0%) | 0.199 | 1 | | | |
| | 25–34 | 132 | 93 (70.5%) | 63 (67.7%) | | 1.40 | 0.79–2.48 | | |
| | 35–44 | 61 | 44 (72.1%) | 34 (77.3%) | | 2.27 | 1.02–5.03 | | |
| | > = 45 | 49 | 39 (79.6%) | 27 (69.2%) | | 1.50 | 0.69–3.26 | | |
| Ethnicity | Dutch | 216 | 159 (73.6%) | 103 (64.8%) | 0.493 | 1 | | | |
| | Other European | 47 | 37 (78.7%) | 29 (78.4%) | | 1.97 | 0.84–4.60 | | |
| | African | 35 | 20 (57.1%) | 12 (60.0%) | | 0.82 | 0.32–2.11 | | |
| | Mid/South American | 65 | 46 (70.7%) | 29 (63.0%) | | 0.93 | 0.47–1.83 | | |
| | Asian | 30 | 22 (73.3%) | 14 (63.6%) | | 0.95 | 0.38–2.41 | | |
| | Other | 9 | 7 (77.8%) | 6 (85.7%) | | 3.26 | 0.38–27.78 | | |
| Educational level | Low | 44 | 30 (68.2%) | 16 (53.3%) | 0.029 | 1 | | 1 | |
| | Mid | 116 | 88 (75.9%) | 52 (59.1%) | | 1.26 | 0.55–2.91 | 1.15 | 0.49–2.72 |
| | High | 216 | 157 (72.7%) | 114 (72.6%) | | 2.32 | 1.04–5.16 | 2.14 | 0.92–4.96 |
| HIV | Negative | 366 | 261 (71.3%) | 168 (64.4%) | 0.047 | 1 | | 1 | |
| | Positive | 35 | 29 (82.9%) | 24 (82.8%) | | 2.66 | 0.98–7.20 | 2.13 | 0.66–6.85 |
| Chlamydia | Negative | 346 | 253 (73.1%) | 168 (66.4%) | 0.941 | 1 | | | |
| | Positive | 56 | 38 (67.8%) | 25 (65.8%) | | 0.97 | 0.47–2.00 | | |
| Gonorrhea | Negative | 362 | 265 (73.2%) | 172 (64.9%) | 0.102 | 1 | | | |
| | Positive | 40 | 26 (65.0%) | 21 (80.8%) | | 2.27 | 0.83–6.22 | | |
| Reported having done sex work in preceding 6 months | No | 373 | 271 (72.7%) | 180 (66.4%) | 0.983 | 1 | | | |
| | Yes | 27 | 18 (66.7%) | 12 (66.7%) | | 1.01 | 0.37–2.78 | | |
| Azithromycin in previous 3 months | No | 393 | 285 (72.5%) | 188 (66.0%) | 0.373 | 1 | | | |
| | Yes | 9 | 6 (66.7%) | 5 (83.3%) | | 2.58 | 0.30–22.39 | | |
| No. of sexual partners in previous 6 months | 0–2 | 112 | 76 (67.9%) | 44 (57.9%) | 0.040 | 1 | | 1 | |
| | 3–10 | 187 | 143 (76.5%) | 93 (65.0%) | | 1.35 | 0.77–2.39 | 1.09 | 0.59–2.02 |
| | >10 | 102 | 71 (69.6%) | 55 (77.5%) | | 2.50 | 1.22–5.13 | 1.91 | 0.81–4.51 |
| Any symptom | No | 297 | 214 (72.1%) | 138 (64.5%) | 0.269 | 1 | | | |
| | Yes | 105 | 77 (73.3%) | 55 (71.4%) | | 1.38 | 0.78–2.43 | | |
| Urogenital discharge | No | 339 | 244 (72.0%) | 161 (66.0%) | 0.780 | 1 | | | |
| | Yes | 63 | 47 (74.6%) | 32 (68.1%) | | 1.10 | 0.56–2.15 | | |
| Dysuria | No | 348 | 252 (72.4%) | 167 (66.3%) | 0.962 | 1 | | | |
| | Yes | 54 | 39 (72.2%) | 26 (66.7%) | | 1.02 | 0.50–2.08 | | |
| Ulcers | No | 396 | 287 (72.5%) | 190 (66.2%) | 0.712 | 1 | | | |
| | Yes | 6 | 4 (66.7%) | 3 (75.0%) | | 1.53 | 0.16–14.93 | | |
| Blood loss | No | 394 | 284 (72.1%) | 189 (66.5%) | 0.603 | 1 | | | |
| | Yes | 8 | 7 (87.5%) | 4 (57.1%) | | 0.67 | 0.15–3.06 | | |
| Pain | No | 398 | 288 (72.4%) | 191 (66.3%) | 0.990 | 1 | | | |
| | Yes | 4 | 3 (75.0%) | 2 (66.7%) | | 1.02 | 0.09–11.36 | | |

MSM: men who have sex with men, OR: Odds ratio, AOR: adjusted odds ratio. P-values, OR and 95%CI are based on clients of whom the samples were typeable.

[1]A patient was considered to be infected with MG-MRAM if in at least one of the samples a mutation was detected. [2]Overall p-value determined with chi-square test.

[3]The multivariable logistic regression model contained the variables Sexual risk group, Educational level, HIV and No. of sexual partners in previous 6 months.

clients were MSM. No association was found between MG-MRAM and co-infections or recent azithromycin treatment. Only 105/402 (26.1%) clients reported symptoms, discharge (63/402, 15.7%) was most frequently reported, followed by dysuria (54/402, 13.4%). There was no significant difference in the proportion reporting symptoms between clients infected with MG-MRAM and clients infected with MG-WT. In multivariable logistic regression none of the included variables was significantly associated with MG-MRAM infections (Table 4). When risk factors were analyzed separately for men and women, the only observed significant association was between MG-MRAM positivity and number of sexual partners in previous 6 months in men (S3 Table).

## Discussion

Here we describe a new qPCR assay with LNA probes to specifically detect MG-WT and MG-MRAM for the most commonly occurring mutations in the 23SrRNA gene of MG. This assay shows a high sensitivity (98.6%) and specificity (94.1%) compared to sequencing analysis. Thus the qPCR with LNA probes can be used as a quick technique to specifically detect MRAM in MG. The MG-MRAM qPCR was able to detect MG in MG-positive samples with comparable frequency as the MgPa PCR, which is often used in routine diagnostics to detect MG [23]. With this new qPCR we found that two-third of the study population was infected with MG-MRAM, thereby likely compromising treatment with azithromycin. Prevalence of MG-MRAM was significantly higher in men, in clients with a higher education, in HIV-positive clients and in clients with >10 sexual partners in the previous six months, but in multivariable analysis no factor was significantly associated with MG-MRAM presence.

The MG-MRAM qPCR was able to detect MG in MG-positive samples as frequently as the MgPa qPCR and there was a good overall agreement between the MgPa qPCR and the MG-MRAM qPCR with a Cohen's kappa of 0.646. However, a substantial portion (37.3%) of the samples that tested positive for MG in the MG-TMA assay could not be typed by the MG-MRAM qPCR. Some samples were positive in the MgPa qPCR whereas they were not typeable with the MG-MRAM qPCR and the other way around. We considered all these samples as correctly detected or typed, since all samples had previously been tested positive with the highly sensitive MG-TMA assay. In most cases, either both assays were able to detect MG or both were not able to detect MG. That a substantial portion of MG positive samples was negative in both qPCR assays can be explained by the superior sensitivity of the MG-TMA assay. In TMA assays RNA molecules are detected, which are present in multiple copies per bacterium. Detection of multiple RNA copies increases the sensitivity compared to detection of the single-copy 23S rRNA gene DNA target in the MG-MRAM qPCR. Earlier studies found between 17.5–40.3% more MG-positives with TMA compared to qPCR [24, 25]. In most studies qPCR is used to detect MG, potentially underestimating the prevalence of MG [1, 11]. Another explanation for the lower detection rates of MG by MG-MRAM qPCR could be that some samples had been stored for more than a year before being tested, while the TMA assay, the MgPa qPCR and sequencing analysis were all performed directly after arrival in the laboratory.

Previously other qPCRs have been described with high sensitivity and specificity that detect MRAM in MG [14–16, 18]. Some of the commercial available qPCRs do not detect MG-WT and can therefore result in incorrectly identified MG-WT when no signal is obtained by the mutation detecting probes [17]. Subsequently, the patient would be treated with a macrolide which, most probably, would lead to treatment failure. Moreover, the RealAccurate TVMGres (Pathofinder) assay does not detect the A2059C mutation which was present in 2.5% of the MG-positive samples in our study according to sequence analysis. Our newly designed

MG-MRAM qPCR detects both MG-WT and MG-MRAM directly and it is a sensitive and specific qPCR for versatile platforms such as the RotorGene which is easy to interpret in routine diagnostics, since there will be only two signals–one for WT and one for MG-MRAM– that need to be analysed.

The MG-MRAM qPCR detected less often the A2058T and A2059C mutations compared to sequencing analysis although this concerned small numbers of samples. This can be explained since the test with the MG-MRAM qPCR mix 2 –containing probes for A2058T, A2058C and A2059C –was often performed more than one year after sequencing analysis, whereas the MG-MRAM qPCR with mix 1 –containing probes for WT, A2058G and A2059G –which was used mostly concomitantly to the sequencing analysis. DNA might possibly already have degraded in these samples. However, our study shows that the most prevalent mutations A2058G and A2059G (Table 2) were detected. Proportions of the mutations were similar to those reported in previous Dutch studies [10, 12]. However, the proportion A2058T seems to be lower in our study population.

We found a very high prevalence of MG-MRAM (66.3%), especially among men, clients with a higher education level, HIV-positive clients and clients with >10 recent sexual partners. In multivariable logistic regression none of these risk factors were significantly associated with MG-MRAM infections. This might be explained by the fact that MSM in this population report a higher education level, are more often HIV-positive and more often have >10 recent sexual partners. The observed prevalence of MG-MRAM is higher than previously reported in the Netherlands (20.9%-44.4%) [10–12]. However, those studies included mainly hospital and primary care clients who were sampled some years ago, whereas our population consisted of clients visiting the STI clinic in urbanized regions in the Netherlands in 2018. In our population we had a relatively large group of MSM and HIV-positives and these groups had a relatively high prevalence of MG-MRAM. Thus treatment with azithromycin is possibly compromised in a majority of MG cases.

This study has some limitations. First of all, we did not sequence all MG positive samples, but only a subset. Second, in our study we only included first test results and client information that was available from that STI clinic visit. We did not have access to information of previous general practitioner or hospital visits, but asked clients if they received antibiotic treatment in the preceding three months. The actual number of clients that received azithromycin in the preceding three months is therefore probably higher. No association between azithromycin treatment in the preceding three months and MG-MRAM was found (Table 4). This might be explained by the low number of clients that reported azithromycin treatment (n = 9).

We found that 8.8% of the clients with a double infection had MG-WT on one anatomical location, and MG-MRAM on the other location. Two of 326 typed samples–from 291 clients– were typed to contain both MG-WT and MG-MRAM, and we classified them as MG-MRAM, since it is to be expected that treatment with azithromycin would not be effective to clear MG in these clients. Infection with both MG-WT and MG-MRAM could be due to infections by different strains, or with the same strain that acquired MRAM on one of the locations. Others reported that MG gains *de novo* resistance mutations–a change from antibiotic-susceptible before treatment to antibiotic-resistant after treatment–in approximately 12% of the cases [26]. If this is true, it is important to test for MG-MRAM in clients at various anatomical locations in order to prescribe the correct treatment if treatment would aim at eliminating MG at possible infection sites, including the anus. However, more research is needed into the spread of azithromycin resistant MG and into the additional value of treating anal infections.

In our view testing solely for the presence of MG is not sufficient to prescribe proper treatment and an additional test for MRAM should be performed. This will help to guide therapy for the individual symptomatic client and prevent further spread of resistant MG. Our newly

developed MG-MRAM qPCR can contribute to this due to its high specificity and sensitivity, and can be used for routine diagnostics.

## Supporting information

**S1 Table.** *Mycoplasma Genitalium* **(MG) detection by different molecular techniques and according to anatomical location.**
(DOCX)

**S2 Table. Comparison of MG-MRAM qPCR and sequencing analysis to detect MRAM in MG.**
(DOCX)

**S3 Table. Prevalence of mutant MG (MG-MRAM) as percentage of typeable samples according to different client characteristics separately for men and women.**
(DOCX)

## Acknowledgments

The authors would like to thank J.M. Brand for his contribution to the study and the technicians of the Public Health Service of Amsterdam and The Hague for assistance with performing all tests.

## Author Contributions

**Conceptualization:** David J. Hetem, Clarissa E. Vergunst, Sophie Kuizenga Wessel, Roel H. T. Nijhuis, Alje P. van Dam, Sylvia M. Bruisten.

**Data curation:** Martijn S. van Rooijen, Maarten F. Schim van der Loeff.

**Formal analysis:** Joyce F. Braam, Maarten F. Schim van der Loeff.

**Investigation:** Joyce F. Braam.

**Methodology:** Joyce F. Braam.

**Supervision:** Alje P. van Dam, Sylvia M. Bruisten.

**Validation:** Joyce F. Braam, Maarten F. Schim van der Loeff.

**Writing – original draft:** Joyce F. Braam.

**Writing – review & editing:** David J. Hetem, Clarissa E. Vergunst, Sophie Kuizenga Wessel, Roel H. T. Nijhuis, Maarten F. Schim van der Loeff, Alje P. van Dam, Sylvia M. Bruisten.

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
