## [Decision Letter · Decision Letter 0]

7 Aug 2020

PONE-D-20-21812

Evaluating the prevalence of and risk factors for macrolide resistance in Mycoplasma genitalium using a newly developed qPCR assay

PLOS ONE

Dear Dr. Braam,

Thank you for submitting your manuscript to PLOS ONE. After careful consideration, we feel that it has merit but does not fully meet PLOS ONE’s publication criteria as it currently stands. Therefore, we invite you to submit a revised version of the manuscript that addresses the points raised during the review process.

We look forward to receiving your revised manuscript.

Kind regards,

Prof Remco PH Peters, MD, PhD, 

Academic Editor

PLOS ONE

Journal Requirements:

3. Thank you for including your funding statement; "For this research we received part of the diagnostic tests to detect MG from Hologic Inc, San Diego, USA."

Reviewers' comments:

Reviewer's Responses to Questions

**Comments to the Author**

1. Is the manuscript technically sound, and do the data support the conclusions?

Reviewer #1: Yes

Reviewer #2: Yes

2. Has the statistical analysis been performed appropriately and rigorously? 

Reviewer #1: Yes

Reviewer #2: Yes

3. Have the authors made all data underlying the findings in their manuscript fully available?

Reviewer #1: Yes

Reviewer #2: Yes

4. Is the manuscript presented in an intelligible fashion and written in standard English?

Reviewer #1: Yes

Reviewer #2: Yes

5. Review Comments to the Author

Reviewer #1: The manuscript reports on the prevalence and risk factors for macrolide resistance in Mycoplasma genitalium using a newly developed qPCR in the Netherlands. Using the new qPCR assay, high rates of MG-MRAM (66.3%) were detected. The manuscript also describes an important new diagnostic PCR assay to detect MG-MRAM in patients.

Major comments

1. The manuscript reports the MG-MRAM qPCR assay has a sensitivity of 98.6% and specificity of 94.1% to detect MG-MRAM compared to sequencing analysis. However, the authors only sequenced a convenience set of 126/309 (41%) MG-MRAM detected by qPCR which is problematic. It would be better if the authors had performed sequencing analysis on all the samples typable with MG-MRAM qPCR instead of the convenience sample set to give a more accurate sensitivity and specify of the new MG-MRAM qPCR assay.

2. The characteristics of the study population including the inclusion/exclusion criteria must be included. This would help clarify the following statements:

• Line 91-92: why anal samples where not collected from all female participants?

• Line 187: how many males and females were recruited?

• Line 190: how many heterosexual males and men who have sex with men were recruited?

3. Table 2: It would be useful to provide a breakdown of the following specimens; urines (MSM and heterosexual males) and anal samples (MSM and women).

4. Limitations of the present study should be indicated.

5.Mentioning Mycoplasma genitalium macrolide resistance associated mutations (MG-MRAM) instead of "mutant MG" throughout the manuscript will help eliminate the confusion since only macrolide resistance associated mutations were investigated in the present study. “Mutant MG” could also suggest other mutations e.g. quinolone resistance associated mutations in MG.

Minor comments

Line 1: Delete “of” the title should read “Evaluating the prevalence and risk factors for macrolide resistance in Mycoplasma genitalium using a newly developed qPCR assay”

Line 73: specify E.coli numbering for nucleotide positions

Line 121: correct to “each primer”

Line 156: correct to “MEGA (version M6.0.6)”

Table 2: column “Hetero male” should read “Heterosexual male”

References: “Mycoplasma genitalium”, “Trichomonas vaginalis”, “Chlamydia trachomatis” and “Neisseria gonorrhoeae” should be in italics

Reviewer #2: The authors described a new qPCR method for detection of macrolide resistant M. genitalium (MG) and used this method to investigate the prevalence and possible risk factors for mutant MG. The new method was validated against the sequencing data and MgPa PCR. Considering the rapidly increasing macrolide resistance in MG, this study provided another new tool to detect the mutant strains and would assist to improve the treatment of MG infections.

The study was based on a big starting sample size (3225). However, the new method was only applied to the “known” positive samples by Hologic MG-TMA assay. If this method is to be used as a “secondary” test just for detection mutations after the Hologic-TMA assay, it is probably fine. If it will be used for MG detection and simultaneous mutant identification, then additional validation on MG negative samples is needed.

Some minor comments:

1. “Clients” was used in most places in this manuscript, while “patients” was also appeared. Whether this is a cultural difference or not, it’s better to keep consistent.

2. Line 108. Please specify the nucleic acid was DNA.

3. Line 151. How to define the “convenience set of 126 samples”? Does it mean random selection?

4. Line 295-299. “That a …MG-MRAM qPCR”. Please rephrase this sentence.

5. Line 330-333. Please discuss/explain the non-significance of the factors after the multivariable logistic regression analysis.

6. PLOS authors have the option to publish the peer review history of their article (what does this mean?). If published, this will include your full peer review and any attached files.

Reviewer #1: No

Reviewer #2: No

---

## [Author Response · Author response to Decision Letter 0]

1 Oct 2020

Amsterdam, September 16th 2020

Re.: Revision PONE-D-20-21812

Dear Professor Peters,

Please find our revised manuscript entitled ‘Evaluating the prevalence and risk factors for macrolide resistance in Mycoplasma genitalium using a newly developed qPCR assay’ by Joyce F. Braam, David J. Hetem, Clarissa E. Vergunst, Sophie Kuizenga Wessel, Martijn S. van Rooijen, Roel H.T. Nijhuis, Maarten F. Schim van der Loeff, Alje P. van Dam and Sylvia M. Bruisten for publication in PLOS ONE. 

We have revised the manuscript in accordance with your comments and those of the reviewers. For convenience we have added a version of our revised manuscript with all changes marked (labelled: Revised Manuscript with Track Changes) as well as a clean version (labelled: Manuscript). Please find below a point-by-point reply to all suggestions. The line-numbers are according to the Revised Manuscript with Track Changes.

On behalf of all authors I would like to thank the reviewers for their critical comments and suggestions and we hope that the revised version of our manuscript is acceptable for publication in PLOS ONE.

Yours Sincerely,

Joyce F. Braam

Public Health Service Amsterdam 

Nieuwe Achtergracht 100

1018 WT Amsterdam

The Netherlands

E-mail: jbraam@ggd.amsterdam.nl

Point-by-point reply to the Journal Requirements:

 and

Response: We indeed used PLOS ONE’s style requirements for the revised manuscript. 

Response: All data were fully anonymized by the clinics’ data managers prior to release to the researchers. The samples that were provided for diagnostic testing could be used for this study; none of these clients had opted out of having their samples stored and possibly re-used after anonymization for research purposes. The Medical Ethics Committee of the Academic Medical Center in Amsterdam (letter reference no. W18.013#18.024) therefore waived the requirement for informed consent of the patients.

This is now also stated more clearly in the manuscript in the paragraph ‘Ethics statement’ (page 12).

3. Thank you for including your funding statement; "For this research we received part of the diagnostic tests to detect MG from Hologic Inc, San Diego, USA."

Response: There are no restrictions on sharing of data or materials and we have added the statement to our revised article. See ‘Funding’, page 26.

Response: We added the statement also in the cover letter. 

Response: There are no competing interests for any of the authors. This was already stated in the Transparency declaration on page 25.

Response: The data that we meant is not a core part of the research being presented in our study, therefore we have now removed the phrase that refers to these data. 

 

Point-by-point reply to the comments of reviewer 1 

Major comments

The manuscript reports on the prevalence and risk factors for macrolide resistance in Mycoplasma genitalium using a newly developed qPCR in the Netherlands. Using the new qPCR assay, high rates of MG-MRAM (66.3%) were detected. The manuscript also describes an important new diagnostic PCR assay to detect MG-MRAM in patients.

1. The manuscript reports the MG-MRAM qPCR assay has a sensitivity of 98.6% and specificity of 94.1% to detect MG-MRAM compared to sequencing analysis. However, the authors only sequenced a convenience set of 126/309 (41%) MG-MRAM detected by qPCR which is problematic. It would be better if the authors had performed sequencing analysis on all the samples typable with MG-MRAM qPCR instead of the convenience sample set to give a more accurate sensitivity and specify of the new MG-MRAM qPCR assay.

Response: We agree with the reviewer that it would have been better if we would have sequenced all MG-MRAM detected by qPCR. However, since we did not make a specific selection of this subset, other than that they were positive in the MG PCR or MG-TMA, we hold that performing sequencing analysis on the remaining samples will not change our results: it would probably only lead to more narrow 95%CIs. Nevertheless, we now added a more specific description on how the set of samples for sequencing analysis was obtained. 

Page 10, paragraph ’Sequencing analysis of 23SrRNA gene’:

“To confirm the mutation qPCR test results, a subset of 126 of the 309 samples (40.8%) that were typable with the MG-MRAM qPCR were used to perform sequencing analysis (Fig 1). In addition, also a subset of 33 MG-positive samples out of the total of 184 samples (17.9%) in which no MG was detected with the MG-MRAM qPCR was used for sequencing analysis (Fig 1), totaling 159 samples for sequencing analysis. For both subsets we selected samples that tested positive with the MgPa qPCR and had a Ct value of <36.”

Furthermore, we have now added the 95%CIs also in the abstract for clarification (lines 39-40). 

2. The characteristics of the study population including the inclusion/exclusion criteria must be included. This would help clarify the following statements:

• Line 91-92: why anal samples where not collected from all female participants?

Response: The routine policy of the STI clinics in The Hague and in Amsterdam are different on collecting anal samples from females and we adhered to these local policies. In The Hague it is standard policy to test all women both vaginal and anal for STI. In Amsterdam women are only tested anally if they are categorized as being at risk for an anal STI (which is defined as: reported anal sex or anal symptoms, were notified for an STI, or reported to perform sex work). For clarification, we rephrased the text as follows: 

Lines 95-98:

“Anal samples were taken from all females attending the STI clinic in The Hague, whereas in Amsterdam anal samples were taken only from females if they reported anal sex or anal symptoms, were notified for an STI, or reported to perform sex work – both according to the local STI clinic policy”.

• Line 187: how many males and females were recruited?

• Line 190: how many heterosexual males and men who have sex with men were recruited?

Response: We agree that the study population was not defined as clearly as possible and we added the following (lines 196-198):

“During the study period 445/3225 (13.8%) tested clients were positive for MG with the MG-TMA assay, of whom 1031 were MSM, 927 heterosexual men, 1249 women, 17 transgender people and the sexual orientation of 2 patients was unknown.”

(lines 199-202)

“From 402/445 clients positive for MG - of whom 182 were MSM, 66 heterosexual men and 154 women - samples were available for MgPa and MG-MRAM testingThe samples included 136 vaginal, 120 urine and 237 anal samples, totaling 493 samples (Fig 1).”

3. Table 2: It would be useful to provide a breakdown of the following specimens; urines (MSM and heterosexual males) and anal samples (MSM and women).

Response: We agree with the reviewer that it is more informative to further specify the specimens according to anatomical locations within hosts and we therefore changed Table 2 as suggested (see page 15). 

4. Limitations of the present study should be indicated.

Response: We agree with the reviewer that the limitations could have been stated more clearly and we have changed the text accordingly. We also added the limitation concerning not sequencing all MgPa positive samples.

Page 24, last paragraph:

“This study has some limitations. First of all, we did not sequence all MG positive samples, but only a subset. Second, in our study we only included first test results and client information that was available from that STI clinic visit.”

5. Mentioning Mycoplasma genitalium macrolide resistance associated mutations (MG-MRAM) instead of "mutant MG" throughout the manuscript will help eliminate the confusion since only macrolide resistance associated mutations were investigated in the present study. “Mutant MG” could also suggest other mutations e.g. quinolone resistance associated mutations in MG.

Response: We agree with the reviewer and consistently replaced ‘mutant MG’ by ‘MG-MRAM’ throughout the manuscript. 

Minor comments

Line 1: Delete “of” the title should read “Evaluating the prevalence and risk factors for macrolide resistance in Mycoplasma genitalium using a newly developed qPCR assay”

Response: Thank you for the suggestion, we changed the title accordingly.

Line 73: specify E. coli numbering for nucleotide positions

Response: We added Escherichia coli numbering to line 73-75 and the sentence now reads (page 5, line 73-75): 

“These tests detect several mutations that are associated with macrolide resistance in the V-region of the 23S rRNA gene: A2058G, A2058T, A2058C, A2059G, and A2059C (Escherichia coli numbering) (10, 19).”

Line 121: correct to “each primer”

Response: We added ‘primer’ to line 124-127 and the sentence now reads: 

“The assay consists of two multiplex qPCRs which use the same forward and reverse primers, 250 nM of each primer and three probes of 125nM each.”

Line 156: correct to “MEGA (version M6.0.6)”

Response: We agree with the reviewer that MEGA should be in capital letters and changed the text accordingly (page 10, line 164) . 

Table 2: column “Hetero male” should read “Heterosexual male”

Response: Table 2 did not contain ‘hetero male’; but tables 3 and 4 did contain this term. We changed in both tables 3 and 4 ‘hetero male’ into ‘heterosexual male’

References: “Mycoplasma genitalium”, “Trichomonas vaginalis”, “Chlamydia trachomatis” and “Neisseria gonorrhoeae” should be in italics

Response: Indeed, as suggested by the reviewer, these species names in the references section are now noted in italics.  

Point-by-point reply to the comments of reviewer 2

The authors described a new qPCR method for detection of macrolide resistant M. genitalium (MG) and used this method to investigate the prevalence and possible risk factors for mutant MG. The new method was validated against the sequencing data and MgPa PCR. Considering the rapidly increasing macrolide resistance in MG, this study provided another new tool to detect the mutant strains and would assist to improve the treatment of MG infections.

The study was based on a big starting sample size (3225). However, the new method was only applied to the “known” positive samples by Hologic MG-TMA assay. If this method is to be used as a “secondary” test just for detection mutations after the Hologic-TMA assay, it is probably fine. If it will be used for MG detection and simultaneous mutant identification, then additional validation on MG negative samples is needed.

Response: We thank the reviewer for these kind words. Indeed, the MG-MRAM qPCR is intended to be used after a sample tests positive for Mycoplasma genitalium (MG) in the Hologic MG-TMA assay but it could also be used in combination with any other assay that detects MG with sufficient sensitivity and specificity.

Minor comments

1. “Clients” was used in most places in this manuscript, while “patients” was also appeared. Whether this is a cultural difference or not, it’s better to keep consistent.

Response: We agree with the reviewer that consistency is important, and accordingly have replaced ‘patients’ by ‘clients’ throughout the manuscript.

2. Line 108. Please specify the nucleic acid was DNA.

Response: We have now specified in line 113 that the nucleic acid was deoxyribonucleic acid and also clarified in lines 114 till 117.

3. Line 151. How to define the “convenience set of 126 samples”? Does it mean random selection?

Response: The sample sets that we used for sequencing analysis have now been described more precisely: 

Page 10, paragraph ’Sequencing analysis of 23SrRNA gene’:

“To confirm the mutation qPCR test results, a subset of 126 of the 309 samples (40.8%) that were typable with the MG-MRAM qPCR were used to perform sequencing analysis (Fig 1). In addition, also a subset of 33 MG-positive samples out of the total of 184 samples (17.9%) in which no MG was detected with the MG-MRAM qPCR was used for sequencing analysis (Fig 1), totaling 159 samples for sequencing analysis. For both subsets we selected samples that tested positive with the MgPa qPCR and had a Ct value of <36.”

4. Line 295-299. “That a …MG-MRAM qPCR”. Please rephrase this sentence.

Response: We agree with the reviewer that lines 295-299 were not clear and replaced the sentences as follows:

“That a substantial portion of MG positive samples was negative in both qPCR assays can be explained by the superior sensitivity of the MG-TMA assay. In TMA assays RNA molecules are detected, which are present in multiple copies per bacterium.” (line 308-311)

5. Line 330-333. Please discuss/explain the non-significance of the factors after the multivariable logistic regression analysis.

Response: We added an explanation of the non-significance of the factors in multivariable logistic regression analysis (page 24):

“We found a very high prevalence of MG-MRAM (66.3%), especially among men, clients with a higher education level, HIV-positive clients and clients with >10 recent sexual partners. In multivariable logistic regression none of these risk factors were significantly associated with MG-MRAM infections. This might be explained by the fact that MSM in this population report a higher education level, are more often HIV-positive and more often have >10 recent sexual partners. The observed prevalence of MG-MRAM is higher than previously reported in the Netherlands (20.9%-44.4%) (10-12). However, those studies included mainly hospital and primary care clients who were sampled some years ago, whereas our population consisted of clients visiting the STI clinic in urbanized regions in the Netherlands in 2018. In our population we had a relatively large group of MSM and HIV-positives and these groups had a relatively high prevalence of MG-MRAM. Thus treatment with azithromycin is possibly compromised in a majority of MG cases.”

---

## [Editor Report · Decision Letter 1]

5 Oct 2020

Evaluating the prevalence and risk factors for macrolide resistance in Mycoplasma genitalium using a newly developed qPCR assay

PONE-D-20-21812R1

Dear Dr. Braam,

We’re pleased to inform you that your manuscript has been judged scientifically suitable for publication and will be formally accepted for publication once it meets all outstanding technical requirements.

Kind regards,

Remco PH Peters, MD, PhD

Academic Editor

PLOS ONE
---

## [Editor Report · Acceptance letter]

7 Oct 2020

PONE-D-20-21812R1 

Evaluating the prevalence and risk factors for macrolide resistance in *Mycoplasma genitalium* using a newly developed qPCR assay 

Dear Dr. Braam:

I'm pleased to inform you that your manuscript has been deemed suitable for publication in PLOS ONE. Congratulations! Your manuscript is now with our production department. 

Kind regards, 

on behalf of

Prof Remco PH Peters 

Academic Editor

PLOS ONE